# GrowLength: Accelerating LLMs Pretraining by Progressively Growing Training Length

## Abstract

The evolving sophistication and intricacies of Large Language Models (LLMs) yield unprecedented advancements, yet they simultaneously demand considerable computational resources and incur significant costs. To alleviate these challenges, this paper introduces a novel, simple, and effective method named "GrowLength" to accelerate the pretraining process of LLMs. Our method progressively increases the training length throughout the pretraining phase, thereby mitigating computational costs and enhancing efficiency. For instance, it begins with a sequence length of 128 and progressively extends to 4096. This approach enables models to process a larger number of tokens within limited time frames, potentially boosting their performance. In other words, the efficiency gain is derived from training with shorter sequences optimizing the utilization of resources. Our extensive experiments with various state-of-the-art LLMs have revealed that models trained using our method not only converge more swiftly but also exhibit superior performance metrics compared to those trained with existing methods. Furthermore, our method for LLMs pretraining acceleration does not require any additional engineering efforts, making it a practical solution in the realm of LLMs.

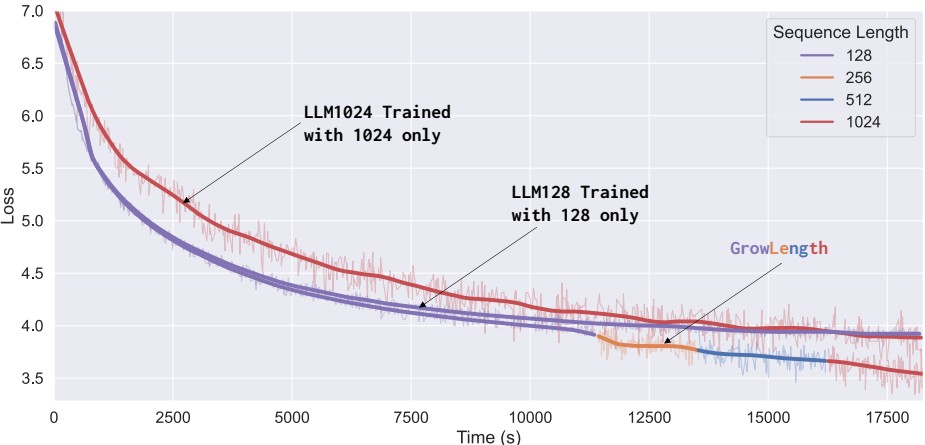

Figure 1: Training curves comparison of our proposed method and the baselines are given the same training time. It shows the training loss curves for Large Language Models (LLMs) trained with fixed sequence lengths of 128 (LLM128), 1024 (LLM1024), and our method. Compared with LLM1024, GrowLength attains a lower loss. This can be attributed to that our method processes more tokens within the same training time, allowing the model to have a broader context. Similarly, the comparison between LLM128 and GrowLength reveals that our method also secures a lower loss in this scenario. This is because, the model trained by our method has experienced longer sequences, enabling better learning ability. Compared with both short or long sequence length instances, our proposed method demonstrates ***enhanced performance within the same pertaining time***, establishing its efficacy over the baseline models.

# 1 INTRODUCTION

The recent surge in the development of Large language models (LLMs) in the realm of natural language processing has dramatically altered the capabilities and applications in the real world (Zhao et al., 2023; Yang et al., 2023). LLMs, with their tremendous parameters and intricate architectures, are facilitating new benchmarks in various tasks that span from text classification to the generation of coherent and contextually relevant narratives. However, the evolution in their capabilities is concomitant with an increased number of model parameters (Li et al., 2023; Touvron et al., 2023a; OpenAI, 2023). LLMs training process demands substantial computational resources with surging costs and has formed obstacles for practitioners and researchers with limited access to such resources.

To mitigate the extensive computational resources and substantial training time requirements, various methods have been introduced to expedite the pretraining of LLMs. Among these, Flash-Attention has been proposed as a solution to accelerate both the training and inference of LLMs (Dao et al., 2022). Quantization-based methods (Dettmers et al., 2022; Liu et al., 2023) seek to reduce the model size by representing model parameters with fewer bits and thus significantly decrease both memory usage and computation cost (Dettmers et al., 2022). Pruning (Ma et al., 2023; Sun et al., 2023) is also commonly adopted to remove redundant model parameters and thus improve efficiency. Existing acceleration methods are essentially model-centric without considering data perspective.

This paper is driven by two key observations: Firstly, LLMs are generally pretrained with relatively short sentences, while many downstream tasks necessitate the processing of long context inputs. This disparity has led to numerous attempts to extend the context window for inference in previous works (Chen et al., 2023; kaiokendev, 2023), demonstrating that content window extension is a feasible solution, requiring only a minimal amount of training with examples during the fine-tuning stage. Secondly, it is well-recognized that training models with shorter sentences are substantially more time-efficient than with longer sequences, a phenomenon explored in detail in Section 3.2. From this understanding, we infer that models trained with shorter sequence lengths can effectively predict long sequences, as evidenced by the success of content window extension. These observations imply that optimizing sentence length during training can lead to more efficient models in the fine-tuning stage. A fundamental question is raised:

*Can content window extension be adapted to the pretraining stage to reduce training time?*

We provide a positive answer for this question via devising a method, named "`GrowLength`", with progressively grow training length during the pre-training of LLM. This method is inspired by the principles of context window extension training paradigms, extending their application to the pretraining phase. Contrary to the fixed sequence length in the pretraining, our proposed method utilizes a dynamic, progressively growing training sentence length. The superiority of this method lies in its adaptability and its capacity to significantly optimize the utilization of computational resources, enabling models to process more tokens in a constrained time frame. Furthermore, by coordinating the training curves with fixed sequence length, we observe considerable improvements in model performance using the same training time, particularly in position extrapolations.

To validate our hypothesis and to demonstrate the superiority of Length Growth over conventional training methods, we designed a series of rigorous experiments encompassing multiple state-of-the-art LLMs. The empirical evidence was compelling. Models nurtured with our approach not only achieved faster convergence rates but also surpassed their counterparts in terms of overall performance metrics. Besides its performance improvement, it also signifies a potential reduction in training costs, a boost in model proficiency, and a democratization of access to high-performance LLMs. We highlight our main contributions as follows:

- We extend the context window extension method to accelerate the pretraining stage of Large Language Models.
- Based on preliminary experiments, we proposed a simple and effective method to accelerate the pretraining of LLMs, without any engineering effort.
- Experimental results demonstrate the effectiveness of our proposed method in the pretraining acceleration of LLMs.

## 2 PRELIMINARIES AND MOTIVATION

In this section, we outline the preliminaries of our proposed method and subsequently present the two key observations that motivate our approach.

### 2.1 POSITIONAL EMBEDDING

In this work, we focus on the Rotary Position Embedding (RoPE) introduced in (Su et al., 2022). RoPE is shown to have excellent position extrapolation ability for context windows extension for instruction tuning (Peng et al., 2023a; Longpre et al., 2023; Gupta et al., 2022). Hereby, we briefly introduce the basic idea of RoPE. We use $w_1, w_2, \cdots, w_L$ to denote a sequence of tokens, and denote their embedding as $\mathbf{x}_1, \cdots, \mathbf{x}_L \in \mathbb{R}^{|D|}$ where $|D|$ is the dimension of the embedding.

The basic idea of RoPE is to impose the positional embeddings into the query and the key vectors. Then, the inner product $\mathbf{q}^T \mathbf{k}$ will contain the relative positional embedding information. The query and key vectors will be transformed as follows

$$\mathbf{q}_m = f_q(\mathbf{x}_m, m) \in \mathbb{R}^{|L|}, \ \mathbf{k}_n = f_k(\mathbf{x}_n, n) \in \mathbb{R}^{|L|}, \tag{1}$$

where $|L|$ is the hidden dimension per head. The functions $f_q, f_k$ are given by

$$f_q(\mathbf{x}_m, m) = W_q \mathbf{x}_m e^{im\theta}, \ f_k(\mathbf{x}_n, n) = W_k \mathbf{x}_n e^{in\theta}, \tag{2}$$

where $\theta_d = b^{-2d/|D|}$, $b = 10000$ and $W_q, W_k : \mathbb{R}^{|D|} \to \mathbb{R}^{|L|}$. The inner product $\mathbf{q}^T \mathbf{k}$ becomes the real part of the inner product $\text{Re}(\mathbf{q}^* \mathbf{k})$. This operation guarantees that the dot product of the query and key vectors will depend solely on the relative distance $m - n$ as follows

$$\langle f_q(\mathbf{x}_m, m), f_k(\mathbf{x}_n, n) \rangle_{\mathbb{R}} = \text{Re}(\langle f_q(\mathbf{x}_m, m), f_k(\mathbf{x}_n, n) \rangle_{\mathbb{C}}) = \text{Re}(\mathbf{x}_m^* W_q^* W_k \mathbf{x}_n e^{i\theta(m-n)}) \tag{3}$$

$$= g(\mathbf{x}_m, \mathbf{x}_n, m - n). \tag{4}$$

The following studies (Rozière et al., 2023; Peng et al., 2023b) indicate that the RoPE possesses the capability to adapt to longer sequences when trained with shorter ones. This advantageous and critical property ensures that progressively increasing the length of the training sequence is viable in our scenario, preventing substantial jumps in loss when transitioning between training sentences of varying lengths.

### 2.2 CONTENT WINDOWS EXTENSION IN FINE-TUNING PHASE

Language models are typically pre-trained with a fixed context length, prompting inquiries into effective methodologies for extending the context length through fine-tuning on relatively smaller datasets. For Large Language Models (LLMs) utilizing positional embedding RoPE, two predominant techniques exist for extending their input length: Direct Position Extrapolation and Position Interpolation (PI). Direct Position Extrapolation enables the model to work on sequences longer than those encountered during the original training. However, its effectiveness diminishes for sequences significantly longer than the pre-trained sequence length. This limitation has led to the development of alternative approaches, notably Position Interpolation (PI), as proposed in works (Chen et al., 2023; kaiokendev, 2023). These studies demonstrate that interpolating the position indices achieves favorable results when fine-tuned with longer sequences, even with a limited number of steps (Chen et al., 2023; kaiokendev, 2023).

Our research is motivated by the principles of applying Direct Position Extrapolation on longer sequences and the advantages offered by fine-tuning with longer sequences. It forms the conceptual foundation upon which we build our methodologies and conduct our explorations, allowing us to critically assess and refine the strategies employed to extend the context length of pre-trained language models.

### 2.3 MOTIVATION

The previous works (Chen et al., 2023; kaiokendev, 2023) have demonstrated that the Content Windows Extension is effective and requires training with only a few examples during the fine-tuning

stage. It is also known that training with shorter sentences consumes significantly less time than training with longer sequences (a detailed analysis is provided in Section 3.2). Based on this knowledge, the following observations can be made:

- Using models trained with shorter sequence lengths has proven to be more effective than training with long sequences, as proven by the Content Window Extension.
- Training with shorter sentences is more time-efficient compared to training with longer sequences.

Motivated by these observations, and considering that the pretraining of LLMs usually requires extensive time, we pose the following question: **Can this paradigm be adapted to the pretraining stage to reduce the pretraining time of LLMs?**

## 3 METHOD

In this section, we introduce our proposed method, `GrowLength`, elucidating its principles and mechanics, derived from the understanding that a dynamic and responsive training timeline can significantly optimize model performance. Given the efficacy of models trained with shorter sequence lengths in predicting longer sequences and their time efficiency, as demonstrated by the Content Windows Extension, we explore adapting this paradigm to the pretraining stage. The aim is to significantly reduce the pretraining time, which is inherently more time-consuming than the fine-tuning stage.

---

**Algorithm 1** PyTorch-style Pseudocode of `GrowLength`

```
# loader_list: data loaders with different
    lengths.
# LLM: language model
# {nubmer}_loader: data loader for text
    sequences with a length of {number}

loader_list = [128_loader, 256_loader,...]

# Train LLMs for N epochs
for loader in loader_list:
    for batch in loader:
        loss = LLM(**batch)
        loss.backward()
        optimizer.step()
```

---

The fundamental concept behind `GrowLength` is that pretraining Large Language Models (LLMs) with shorter sequences is substantially faster than training with longer sequences. Additionally, transitioning from shorter to longer sequences does not induce a loss jump and preserves the degradation trend. In this approach, the pretraining phase begins with shorter sequences and progressively extends the sequence length as training advances. Our proposed method is straightforward, and we present the pseudocode for our method below.

### 3.1 IMPLEMENTATION

In this section, we outline two key methods of implementation: Positional Extrapolation and Positional Interpolation.

- **Positional Extrapolation** This method involves estimating unknown positions utilizing known positions in the sequence. It is particularly beneficial for generating predictions outside the range of available data points. This method usually works well when the length of the long text is close to the training length.
- **Positional Interpolation** Positional Interpolation, on the other hand, is the method of estimating unknown positions by using two or more known positions within the range of available data points. This method is crucial for generating accurate and reliable predictions within the known range, enabling us to fill in the gaps in our knowledge with confidence.

Based on our experiments, we noticed the direct positional extrapolation works quite well in our method, as shown in Figure 1. Thus in our implementation, we adopt direct positional extrapolation.

### 3.2 WHAT ADVANTAGES CAN BE GAINED BY TRAINING LLMS WITH SHORTER SEQUENCES?

In this section, we experimentally analyze the computational complexity of Large Language Models (LLMs) with varying lengths of sequence[1]. Three primary factors under consideration are running time, memory usage, and the number of tokens processed. We have summarized the results in the tables below. Through this analysis, we infer that the computational complexity of LLMs is heavily

---

[1]The experiments are conducted on a single A100-80G GPU.

influenced by the length of the sequence, impacting the running time, memory usage, and the number of tokens processed by the models.

Table 1: Comparison of the running time of Large Language Models (LLMs) with different sequence lengths. The row "Running time ratio" is computed by normalizing each running time value by the running time observed for a sequence length of 16384.

| Length of Sequence | 128 | 256 | 512 | 1024 | 2048 | 4096 | 8192 | 16384 |
|---|---|---|---|---|---|---|---|---|
| Running Time (s) | 0.18 | 0.18 | 0.19 | 0.22 | 0.30 | 0.27 | 0.39 | 0.60 |
| Running Time Ratio | 30% | 30% | 32% | 37% | 49% | 44% | 66% | 100% |

Table 2: The comparison of the memory of LLMs with different lengths of sequence. In this experiment, we measure the memory usage for one training step across varying sequence lengths, ranging from 128 to 16384. The row "Memory Usage Ratio" is computed by normalizing each running time value by the running time observed for a sequence length of 16384.

| Length of Sequence | 128 | 256 | 512 | 1024 | 2048 | 4096 | 8192 | 16384 |
|---|---|---|---|---|---|---|---|---|
| Memory Usage (GB) | 9.49 | 10.06 | 11.19 | 13.45 | 17.95 | 17.95 | 22.35 | 41.63 |
| Memory Usage Ratio | 30% | 30% | 32% | 37% | 49% | 44% | 66% | 100% |

Table 3: Comparison of the total number of tokens accommodated while utilizing the full capacity of the GPU's available memory. In this experiment, we assess the number of tokens processed, with sequence lengths varying from 128 to 16384, exploiting the entire available memory of the GPU. The "Num of Tokens Ratio" row is computed by normalizing each value against the total number of tokens counted for a sequence length of 16384.

| Length of Sequence | 128 | 256 | 512 | 1024 | 2048 | 4096 | 8192 | 16384 |
|---|---|---|---|---|---|---|---|---|
| Num of Tokens | 97010 | 91621 | 89043 | 69228 | 52663 | 54805 | 33211 | 28248 |
| Num of Tokens Ratio | 343% | 324% | 315% | 245% | 186% | 194% | 118% | 100% |

Table 1 shows that, as expected, the running time for one training step increases with the increase in sequence length. Specifically, the running time is the lowest (0.18 seconds) for a sequence length of 128 and the highest (0.60 seconds) for a sequence length of 16384. We conclude that **training LLMs with shorter sequences is much faster than training with longer sequences.**[2].

Table 2 shows that the memory usage significantly increases with the increase in sequence length. For a sequence length of 128, the memory usage is 9.49 GB, while it escalates to 41.63 GB for a sequence length of 16384. We conclude that **when consuming the same GPU memory, training with shorter sequences allows the use of a larger batch size.**

Table 3 shows that the total number of tokens accommodated decreases with the increase in sequence length when utilizing the full capacity of the GPU's available memory. We conclude that **for smaller sequence lengths, the model can process a higher number of tokens simultaneously, exploiting the entire available memory of the GPU**.

### 3.3 DISCUSSION

We discuss how our proposed methods relate to other techniques and their implications.

- **Orthogonal to Other LLM Acceleration Methods** Our proposed method is distinct and orthogonal to other Large Language Model (LLM) acceleration techniques, implying that it can be integrated with them without causing redundancy. This unique characteristic enables the enhancement of the efficiency and effectiveness of existing acceleration methods through the incorporation of our approach, offering new dimensions for exploration in the acceleration of LLMs.

---

[2]In the experiment for Table 1 and Table 2, we keep the total number of tokens in one batch same for different sequence length. Each batch contains 16384 tokens

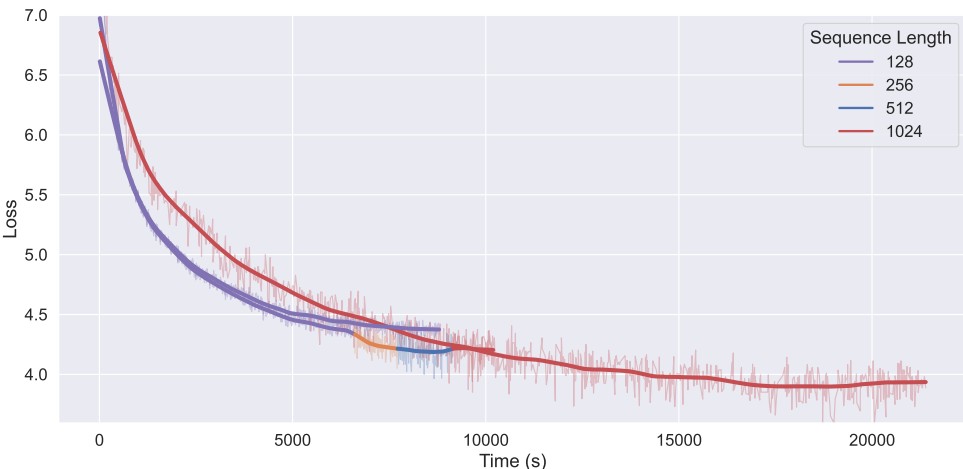

Figure 2: Comparison of the LLMs trained with the same total of tokens.

The orthogonality of our proposed method allows for its seamless integration with existing acceleration strategies, thereby opening new pathways for exploring synergies and augmenting the overall efficiency and effectiveness of Large Language Models.

- **Observe More Tokens for Enhanced Performance** It is obvious that examining more tokens can significantly enhance the model's comprehension and performance. Our method is especially proficient in this regard, being able to process more tokens quickly. Given the accelerated token processing capability of our method, it serves as an efficient strategy to expedite the preliminary stages of training for LLMs. By assimilating a greater number of tokens in a limited time, the model can gain additional advantages, enabling it to make more precise and informed predictions.

## 4    EXPERIMENTS

In this section, we conduct experiments to demonstrate the effectiveness of our proposed method. The experiments, by default, are conducted using a 160M LLM. All models utilized in our experiments adopt the consistent configurations as the Pythia model (Biderman et al., 2023), albeit with varying sizes.

### 4.1    HOW FAST CAN THE PROPOSED METHOD ACCELERATE THE LLMS PRETRAINING?

In this subsection, we perform experiments to evaluate the efficiency of our proposed method in accelerating the pretraining of LLMs. We vary the training sequence length to train LLMs and our proposed method, and then we compare the running time associated with different lengths. To maintain a fair comparison, we ensure that the total number of tokens used for training remains constant across different settings. Our experimental setups for LLM pretraining are as follows:

- LLM128: In this setting, we utilize a fixed training sentence length of 128 tokens, totaling 0.36B tokens across all sentences.
- LLM1024: This setup involves a fixed training sentence length of 1024 tokens, maintaining the same total number of tokens as in LLM128, allowing for a direct comparison of running times at different sequence lengths.
- `GrowLength`: We employ our method, which trains the LLMs starting from 128 tokens and progressively grows to 1024 tokens. Pretraining with a length of 128 significantly saves time, and the final stage of pretraining at a length of 1024 enhances the performance of LLMs.

From Figure 2, we have the following two main observations: Firstly, when maintaining an equivalent count of tokens, LLM1024 requires a longer pretraining duration in comparison to LLM128. This training time is also increased with the growed computational requisites using longer sequence lengths. Secondly, in contrast to LLM1024, our method requires significantly less time for training,

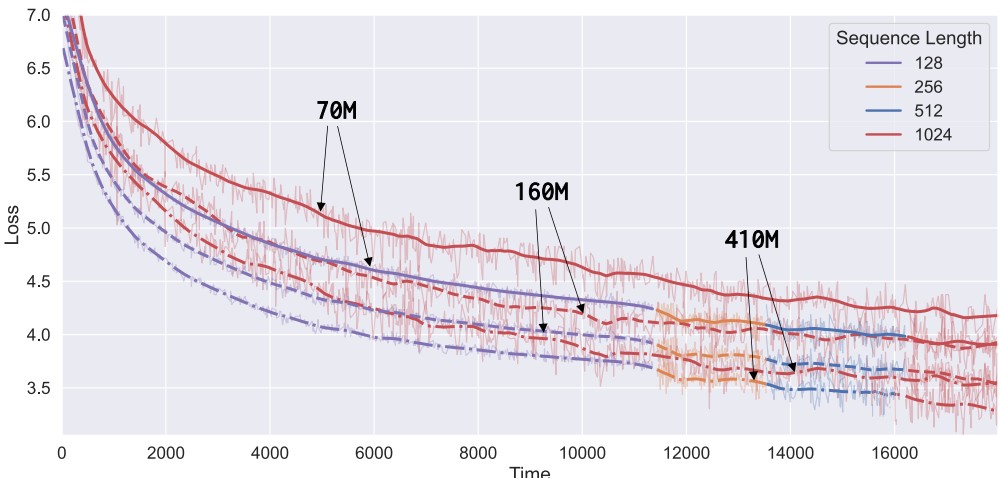

Figure 3: Comparison of the different sizes of models w/ and w/o **GrowLength**. Three model pairs (70M, 160M, 410M) are trained at the same time.

demonstrating superior computational efficiency for the same number of tokens. Last, when compared to LLM128, **GrowLength** exhibits a lower loss, indicating a more powerful and efficacious training. These findings highlight **GrowLength**'s capability in terms of computational efficiency, and the practical value in resources-constrained scenarios.

### 4.2 WILL THE PROPOSED METHOD RESULT IN THE SAME OR LOWER LOSS ?

In this section, we conduct a series of experiments to assess the effectiveness and efficiency of our proposed method. We aimed to understand whether our approach could reach the same loss as that of baselines when LLMs are trained within the same time span. We systematically compared our proposed method against standard approaches under identical settings, primarily focusing on the training of LLMs within the same time span. This comparative analysis allowed us to observe the inherent advantages and potential limitations of our method, providing a comprehensive understanding of its practical implications. The LLMs were trained using the same datasets, and the training parameters were kept consistent across different models to avoid any discrepancies in the results. Each model's performance was evaluated based on the loss. The results are shown in Figure 1.

It shows the training loss curves for LLMs trained with fixed sequence lengths of 128 (LLM128), 1024 (LLM1024), and our method. Compared with LLM1024, **GrowLength** attains a lower loss. This can be attributed to the ability that process more tokens within the specific training time, allowing the model to have a broader context. Similarly, the comparison between LLM128 and **GrowLength** reveals that our method also secures a lower loss in this scenario. This is because, the model trained by our method has experienced longer sequences, enabling better learning ability. In both short or long sequence length instances, our proposed method demonstrates enhanced performance within the same pertaining time, establishing its efficacy over the baseline models.

### 4.3 HOW DOES OUR PROPOSED METHOD PERFORM ON DIFFERENT SIZES OF THE LLMS?

In this section, we evaluate the efficiency of our proposed methods across LLMs of diverse scales, specifically focusing on different LLMs with 70M, 160M, and 410M parameters. Our assessment involves various sizes of models, including 70M, 160M, and 410M. Experiments utilize the most conducive training schedules, determined by preliminary analyses, ensuring optimal conditions for each model size and mitigating potential biases in our assessments.

**Results.** From Figure 3, we can obtain two observations: firstly, while maintaining an equivalent length of time, **GrowLength** can consistently obtain lower loss across the three different sizes of models. This observation suggests that **GrowLength** can scale up to larger LLMs effectively. Second,

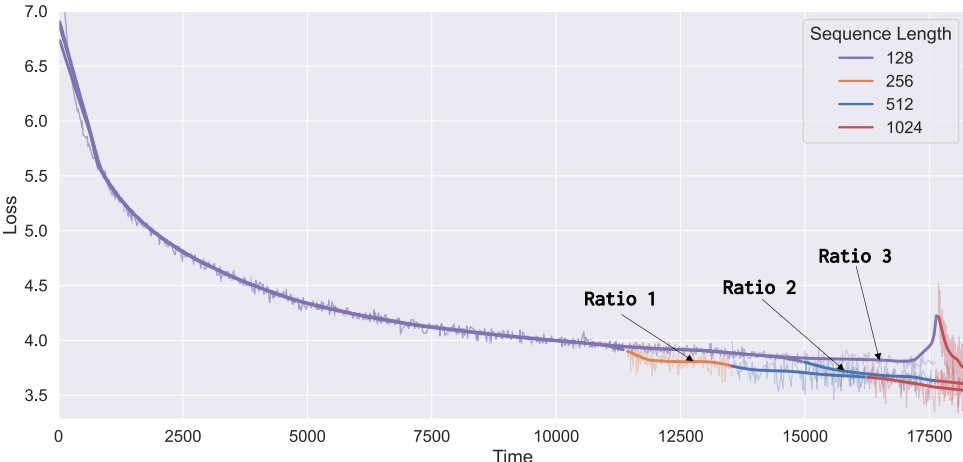

Figure 5: Comparison of the different ratios of the training length. There are three different ratios: 128, 256, 512, 1024; 128, 512, 1024; 128, 102.

our **GrowLength** does not influence the scaling property of LLMs. We can see, that the smaller model still has a higher loss compared to the larger model with the same training time. In the meantime, we can notice that, with the help of **GrowLength**, the smaller model can achieve a very close loss to the larger model with normal pretraining. For example, in this figure, at the end of the training, the 70M model with **GrowLength** reached the same or even slightly lower loss compared to the 160M model.

## 4.4 WILL OUR METHODS SHOW BETTER CONTEXT WINDOWS EXTENSION ABILITIES?

In this section, we delve into a comparative analysis to determine whether our proposed method exhibits enhanced context window extension abilities. compared the baselines. For the long evaluation text, we utilized the dataset used by (Peng et al., 2023b), ensuring that we had diverse and representative samples to validate the robustness and versatility of our methods. Within our experimental setting, **GrowLength**-1, LLM128, and LLM1024 are trained with the same number of tokens, while **GrowLength**-1 is trained with more tokens.

When comparing **GrowLength**-1, LLM1024, and LLM128, **GrowLength**-1 consistently outperforms the others across all input sizes, illustrating its su-

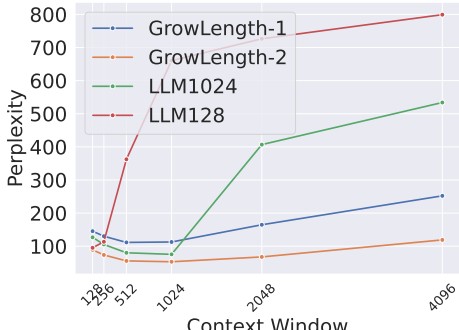

Figure 4: Comparison of the context window extension abilities

periority among all the baselines. LLM128 displays significant deterioration, especially with larger input sizes, highlighting potential limitations in scalability. **GrowLength**-2 provides a more stable performance since it was trained with more tokens. This concise analysis underscores the effectiveness of our proposed methods in extending context windows.

## 4.5 THE INFLUENCE FROM RATIOS OF DIFFERENT WINDOW SIZE DURING TRAINING

In this subsection, we delve deeper into understanding how varying the ratio of sequence lengths impacts the efficiency and efficacy of LLMs pretraining. Our primary goal is to examine whether there is a significant difference in pretraining efficiency and model performance when using different sequence lengths, and if so, we can adjust the ratio of different training lengths for pertaining. For example, we can enlarge the ratio of the shorter sequence to future accelerate the pretraining.

To control the ratio of length 128, we adjust the ratio of training time with different context window sizes. This allows us to systematically study the influence of sequence length on model training dynamics and final performance. We didn't pay effort in the ratio selection and we heuristically selected the ratio in each setting.

Our observations from the controlled experiments reveal two key insights:

- **GrowLength** is not sensitive to the ratio of different window size. For either w/ or w/o the **256** window size during pretraining, the model can reach almost the same time at the end of training. The loss transition is smooth. [Ratio1& Ratio2]
- However, the substantial difference between consecutive training window sizes in **GrowLength** can lead to dramatic loss rising and a drop in performance [Ratio3]

## 5 RELATED WORKS

This section introduces two lines of work that are related to our method, namely, Efficient LLMs and positional encodings within LLMs.

**Efficient LLMs.** There has been increasing interest in developing an efficient method for pretraining large language models (LLMs) Kim et al. (2023). Dao et al. (2022); Choi et al. (2022); Kwon et al. (2023) optimize the CUDA kernels to reduce memory access and improve both training and inference speed. Approaches involving pipeline parallelism Shoeybi et al. (2019); Huang et al. (2019) and tensor parallelism Shoeybi et al. (2019); Li et al. (2021) have facilitated the distribution of workload across multiple GPUs, enhancing the efficiency of scaling LLM inference. Quantization, another pivotal method, has been explored for compressing LLM parameters to optimize inference efficiency Wu et al. (2023); Dettmers et al. (2022); Frantar et al. (2022). In the development of new LLMs, managing computational costs and time is crucial, making these advancements paramount for progress in the field. Our method is orthogonal to existing methods and can be integrated with them to further enhance training acceleration.

**Positional Encodings in LLMs.** Various transformer architectures typically incorporate position information, e.g., positional encodings (Vaswani et al., 2017; Black et al., 2022; Penedo et al., 2023; Kazemnejad et al., 2023). The initial design of positional embedding is *absolute positional encodings*, which are learnable position embeddings (Kenton & Toutanova, 2019) that provide the absolute positions. Then, sinusoidal position embeddings, fixed position embeddings, encode the token positional information embeddings (Vaswani et al., 2017). Subsequently, a learnable or fixed bias is proposed to be added to the dot product between two tokens' position embeddings on the attention score (Ke et al., 2020). The modern LLMs architecture usually adopts *relative positional encodings*, which only use distance information between tokens instead. The relative embedding usually adds a learnable bias to the attention score (Raffel et al., 2020; Dai et al., 2019). Alibi Press et al. (2021) proposes adding a fixed linear attention bias. Then, Su et al. (2022) creatively proposes rotating positive embedding RoPE (Su et al., 2022; Touvron et al., 2023a; Rozière et al., 2023; Touvron et al., 2023b), and XPos (Sun et al., 2022) extends the RoPE for extrapolation ability.

## 6 CONCLUSION AND IMPACT

We propose the **GrowLength** method aimed at accelerating the pretraining of Large Language Models (LLMs) by progressively increasing the training length. Given that the pretraining phase consumes the majority of the training time for LLMs, and considering the recent successes in extending the context window during fine-tuning of pretrained LLMs, we believe that expanding the context windows during the training of LLMs holds significant promise. Motivated by these observations, we extend and adopt the context window extension technique to the pretraining stage to reduce the overall pretraining time. Our method allows LLMs to process more tokens using shorter sequence lengths in the initial stages of training. We conducted experiments to demonstrate the effectiveness of our method. To the best of our knowledge, our paper is the first work that accelerates LLM pretraining from the input sequence perspective and is compatible with existing acceleration methods.

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
