# OpenReview forum: "GrowLength: Accelerating LLMs Pretraining by Progressively Growing Training Length"
_ICLR.cc/2024/Conference — Submitted to ICLR 2024_

### Official Review · Reviewer_gxVT · 2023-10-12

**Soundness:** 2 fair
**Presentation:** 2 fair
**Contribution:** 2 fair
**Rating:** 3
**Confidence:** 4

**Summary:**

> **TL;DR:** The proposed GrowLength method progressively increases the LLM training length throughout the pre-training phase, thereby mitigating computational costs and enhancing efficiency. However, I find the paper lacking comparison to the common BERT two phase pre-training approach which increases the context window in the second phase. Addressing my concerns and questions would improve my score, specifically W.1 and W.2.

The paper proposes the GrowLength method to reduce the computational cost of training LLMs. The high computational cost of LLMs is an ongoing challenge with plenty of recent research discoveries. Contrary to the fixed sequence length in the pretraining, the proposed GrowLength method utilizes a dynamic, progressively growing training sentence length. The superiority of this method lies in its adaptability and its capacity to significantly optimize the utilization of computational resources, enabling models to process more tokens in a constrained time frame.

**Strengths:**

* **S.1.** The proposed GrowLength algorithm tackles an important problem in the computational costs of training LLMs.
* **S.2.** The experiments show that the GrowLength method outperforms the common constant context length approach.
* **S.3.** The paper provides results on models of different sizes.

**Weaknesses:**

* **W.1.** The paper lacks comparison to the BERT [1] pre-training, which used a two-step pre-trianing approach with a growing context window length.
* **W.2.** The figures are confusing with different arrows pointing at the lines with same colors (Figure 3 & 5).
* **W.3.** The experiments are conducted on a single neural architecture and the provided architecture sizes are considerably small compared to existing LLMs.

[1] Devlin, Jacob, Ming-Wei Chang, Kenton Lee, and Kristina Toutanova. "Bert: Pre-training of deep bidirectional transformers for language understanding." arXiv preprint arXiv:1810.04805 (2018).

**Questions:**

* **Q.1.** Where are GrowLength-1 and GrowLength-2 defined?
* **Q.2.** How would the GrowLength method work with substantially larger model 7B+?
* **Q.3.** How would the GrowLength method work with substantially context windows?

---

> ### Author Response · Authors · 2023-11-22
> **Response to Reviewer gxVT**
>
> Dear Reviewer gxVT,
>
> We sincerely thank you for your time and effort in reviewing our paper. We value your suggestions and address them point by point in the following:
>
>
> **Q.1. The paper lacks comparison to the BERT [1] pre-training, which used a two-step pre-trianing approach with a growing context window length.**
> The pre-training methods of BERT and GPT-NeoX are distinct. BERT employs a two-step approach with an expanding context window, focusing on masked language modeling and next sentence prediction. In contrast, GPT-NeoX utilizes an autoregressive training strategy. Considering these differences, we did not choose the BERT pre-training as a baseline.
>
>
>
> **Q.2. The figures are confusing with different arrows pointing at the lines with same colors (Figure 3 & 5).**
> Thanks for your suggestion, we modified the figure to make it clearer.
>
>
> **Q.3. The experiments are conducted on a single neural architecture and the provided architecture sizes are considerably small compared to existing LLMs.....How would the GrowLength method work with substantially larger model 7B+?**
>
> Larger models, such as those with 7B+ parameters, require significantly more computational resources. Implementing the GrowLength method at such scales could be prohibitively resource-intensive in terms of computational power and energy consumption. This limitation could affect the practicality of using the method in real-world applications, where resource constraints are a significant consideration. Moreover, it is common practice to use smaller models to verify experimental settings, as seen in the Llama2 paper. In this paper, the GQA setting is verified using a smaller model (30B) before being applied to a larger model (70B).
>
> Also, it has been found that much smaller models' performance can predict the performance of huge models. This has been discussed in GPT-4 technical report[1], PaLM-2 report[2] and Chinchila[3]. For examples, by the figure 1 and figure 2 from GPT-4's report, much smaller models (1000x smaller) can be used to predict GPT-4's performance. Hence, we kindly argue that, the lack of results from pretty large models does not harm the soundness of our mehtods.
>
> [1][GPT-4 Technical Report](https://arxiv.org/pdf/2303.08774.pdf)
> [2][PaLM 2 Technical Report](https://arxiv.org/pdf/2305.10403.pdf)
> [3][Training Compute-Optimal Large Language Models(Chinchilla)](https://arxiv.org/pdf/2203.15556.pdf)
>
>
> **Q.4. Where are GrowLength-1 and GrowLength-2 defined?**
>
> In Figure 4, 'GrowLength-1' represents the model trained using our growth strategy. In contrast, 'GrowLength-2' employs the same strategy but processes approximately twice as many tokens as 'GrowLength-1'. 'GrowLength-1' demonstrates accelerated training times compared to models using a fixed context length of 1024, as shown in Figure 2.
>
> When processing more tokens, the training duration of 'GrowLength-2' is comparable to that of a fixed-length model, namely 'LLM1024'. We have included both models in the figure to ensure a comprehensive and fair comparison.
>
>
>
>
> **Q.5. How would the GrowLength method work with substantially context windows?**
> **if we understand correctly, the reviewer want the performance of LLM with longer context windows**
>
> The LLM pre-trained by our proposed method consistently demonstrates superiority over all the baselines when comparing GrowLength-1, LLM1024, and LLM128. The results are presented in Figure 4.
>
>
> Your feedback is invaluable to us, and we trust that our response has effectively addressed your concerns. In light of this, we would be grateful if the reviewer could reassess our work and possibly reconsider the score given.
>
> Thanks,\
> Authors

---

### Official Review · Reviewer_w2Jc · 2023-10-30

**Soundness:** 3 good
**Presentation:** 3 good
**Contribution:** 2 fair
**Rating:** 5
**Confidence:** 4

**Summary:**

This paper proposed a simple but effective training strategy for LLMs, namely GrowLength, which changes the data loader, by progressively feeding longer training data during the pre-training phase. In this way, the model training process can be accelerated. GrowLength is motivated by the context windows extension methods for fine-tuning, which indicates that model trained with shorter sequence can also benefit tasks with longer inputs, and employ the direct positional extrapolation for implementation. Many experiments are presented in this paper, showing that the motivation is reasonable, and the proposed GrowLength is effective.

**Strengths:**

- This paper targets a little-explored aspect for LLMs, which is adjusting training data to accelerate training process. Since the training costs for LLMs are huge, I think this paper targets a very important research question.
- The presentation in this paper is clear. This paper is well-written, and results are clearly demonstrated with figures or tables.
- Plenty of experiments in this paper make the proposed idea convincing. The motivation of the paper comes from some experimental observations, regarding the computational complexity of LLMs with varying lengths of sequences. And the proposed GrowLength is evaluated and analyzed from multiple aspects, including training time, training loss, model size, and so on.

**Weaknesses:**

- Some work related with general language processing but not LLMs is not discussed/compared, such as Curriculum Learning for Natural Language Understanding (https://aclanthology.org/2020.acl-main.542.pdf ). The proposed method is quite related to curriculum learning, where an easy-to-difficult curriculum is arranged for model training. This paper also has a baseline, which uses question length/paragraph length as difficulty metrics.
- How to determine the growing context window size during training (like 128, 256, 512, …) is not rigorously studies. Will this exponential growth be too fast, especially for the latter stage, or even longer sequence (e.g., 1B tokens)? And maybe direct positional extrapolation will not work well when growing too fast.
- The unique challenge when applying content window extension to pretraining stage is not very clear. It seems that simply using the technique proposed for fine-tuning also works well.
- No final testing results are presented. Although this paper is working on optimization but not generalization. But final performance on testing set of various tasks is the thing that matters a lot. Besides loss, these results should also be included and analyzed.

**Questions:**

- Will this method be sensitive to the length distribution of the training data? If the pre-training dataset has few samples with short sequences, will GrowLength still be effective?
- The first observation presented before section 3 is not very rigor. Context window extension methods are proposed because LLMs are pre-trained with a fixed context window, but we would like to apply them to tasks with longer sequences. So, these methods just tell us that context length can be extended, but not say “trained with shorter sequence lengths has proven to be more effective than training with long sequences”. Not sure what the author is trying to say here.
- What LLM is used in all experiments? I am curious about how this method works in experiments when applied to LLMs which already have some techniques for accelerating and handling long context, such as GQA in llama2 and FastAttention, given that the authors have explained that they are orthogonal, but experiments are more convincing here.
- Table 1 2 3 might be better presented with figures.
- Typos: At the end of the 3rd line in Sec. 4.4, the full stop should be removed. Also the final sentence in this paragraph, the latter one should be GrowLength-2.

---

> ### Author Response · Authors · 2023-11-22
> **[1/2] Response to Reviewer w2Jc**
>
> Dear Reviewer w2Jc,
>
> We sincerely thank you for your time and effort in reviewing our paper. We value your suggestions and address them point by point in the following:
>
>
> **Q1: How to determine the growing context window size during training (like 128, 256, 512, …) is not rigorously studies. Will this exponential growth be too fast, especially for the latter stage, or even longer sequence (e.g., 1B tokens)? And maybe direct positional extrapolation will not work well when growing too fast.**
>
> Our growth schedule was determined empirically. We agree that significant discrepancies in stage lengths could impair performance during pretraining, as indicated in Figure 5. However, our experiments suggest that, at least in our current setting, exponential growth does not negatively affect our results. Regarding super-large context window sizes, the proposed methods may not be effective.
>
> Some existing studies [1,2] have extended the context length by fine-tuning, for example, scaling Llama-2 from 4k to 8k or even 32k. This supports the viability of our method. Given the abundance of data in the pretraining phase compared to fine-tuning, we believe that a growth strategy could be effective even with larger length variations between stages.
>
> It is also important to note that we do not claim our schedule is optimal. The key innovation of our research is the introduction of increasing context lengths during pretraining, which we believe enhances the process.
>
> [1][Extending Context Window of Large Language Models via Positional Interpolation](https://arxiv.org/abs/2306.15595)
> [2][LongLoRA: Efficient Fine-tuning of Long-Context Large Language Models](https://arxiv.org/abs/2309.12307)
>
> **Q2:The unique challenge when applying content window extension to pretraining stage is not very clear. It seems that simply using the technique proposed for fine-tuning also works well.**
>
> Our approach aims at reducing pretraining time for models with a fixed sequence length, such as 1024 tokens. While our method focuses on minimizing pretraining duration, we recognize that our fine-tuning technique is still applicable and beneficial for various purposes during the fine-tuning stage.
>
>
> **Q3: No final testing results are presented. Although this paper is working on optimization but not generalization. But final performance on testing set of various tasks is the thing that matters a lot. Besides loss, these results should also be included and analyzed.**
>
> We conducted experiments to investigate the performance comparsion on other downstream tasks. Specifically, we ran several tasks using [lm-evaluation-harness](https://github.com/EleutherAI/lm-evaluation-harness/tree/master). The results show that our proposed method outperforms the baseline in 7 out of 9 tasks.
>
> | Dataset | Staged Training | Fixed Length Training(1024) |
> |---------|---------|---------|
> | boolq   | **0.4465**  | 0.3804  |
> | multirc | **0.0189**  | 0.0105  |
> | rte     | 0.5126  | **0.5235**  |
> | wic     | **0.5235**  | 0.5047  |
> | wnli    | **0.4507**  | 0.4366  |
> | cb      | **0.5179**  | 0.4107  |
> | qasper  | **0.3356**  | 0.0     |
> | webqs   | **0.0009**  | 0.0005  |
> | wsc     | 0.3654  | 0.3654  |
>
>
>
> **Q4: Will this method be sensitive to the length distribution of the training data? If the pre-training dataset has few samples with short sequences, will GrowLength still be effective?**
>
> Our GrowLength method remains effective even when the pre-training dataset contains a limited number of short sequences. Additionally, the nature of pre-training data for Large Language Models (LLMs) typically involves large volumes, allowing for the selective inclusion of pre-training text data. It is important to highlight that our proposed pre-training approach demonstrates superior generalization capabilities for longer texts, as illustrated in Figure 4. This indicates that our method is robust against variations in test text length.

---

> > ### Author Response · Authors · 2023-11-22
> > **[2/2] Response to Reviewer w2Jc**
> >
> > **Q5: What LLM is used in all experiments? I am curious about how this method works in experiments when applied to LLMs which already have some techniques for accelerating and handling long context, such as GQA in llama2 and FastAttention, given that the authors have explained that they are orthogonal, but experiments are more convincing here.**
> >
> > In our experiments, we employed the Pythia model([Pythia: A Suite for Analyzing Large Language Models Across Training and Scaling](https://arxiv.org/pdf/2304.01373.pdf)]. Our proposed training strategy is designed to be model-agnostic by altering only the input sequence length, which does not interfere with the core mechanisms of transformer models.
> >
> > Specifically, the GQA technique in Llama2 operates at the architectural level and is independent of the input sequence length. Meanwhile, FastAttention functions at the computational level, serving as an equivalent to the traditional self-attention mechanism.
> >
> > We acknowledge the value of further experimental validation. However, due to computational resource constraints, we were unable to include such results in our study.
> >
> >
> > **Q6: Table 1 2 3 might be better presented with figures.**
> >
> > We appreciate your suggestion to enhance the presentation of Table 1, Table 2, and Table 3. In response, we have incorporated these elements into the updated manuscript as figures to facilitate better understanding and visual appeal.
> >
> >
> >
> > Your feedback is invaluable to us, and we trust that our response has effectively addressed your concerns. In light of this, we would be grateful if the reviewer could reassess our work and possibly reconsider the score given.
> >
> > Thanks,\
> > Authors

---

### Official Review · Reviewer_jdbk · 2023-10-30

**Soundness:** 1 poor
**Presentation:** 3 good
**Contribution:** 2 fair
**Rating:** 3
**Confidence:** 4

**Summary:**

This paper proposes to progressively increase input length to accelerate training. By utilizing the functionality of RoPE embedding, the longer sequence is able to adapt the model trained with shorter sequences.

**Strengths:**

- Good writing;
- The experiments show that progressively growing sequence length can accelerate the training process.

**Weaknesses:**

- My largest concern lies in the validation of the experiment. The model is trained around 20000s (5.56h) at most, which is too short to validate the pretraining process since the model is far away from convergence. Notice that [1] trains model for 300B tokens, and a 160M LLM usually needs several days to converge on 16 V100s, to my knowledge. I completely understand that resource demand is high for researchers; however, it is hard for me to agree with the conclusion from the present experiment.
- A shorter sequence surely leads to a smaller computation complexity. Therefore, this insight is not really original.
- Typos:
	- "while GrowLength-1 is trained with more tokens." -> while GrowLength-2 is trained with more tokens;
	- Lacks the conference of [1] in the paper;
	- etc.


[1] Stella Biderman, Hailey Schoelkopf, Quentin Anthony, Herbie Bradley, Kyle O’Brien, Eric Hal- lahan, Mohammad Aflah Khan, Shivanshu Purohit, USVSN Sai Prashanth, Edward Raff, Aviya Skowron, Lintang Sutawika, and Oskar van der Wal. Pythia: A suite for analyzing large language models across training and scaling, ICML 2023.

**Questions:**

- In Sec 4.4, how many extra tokens are used by GrowLength-2?

---

> ### Author Response · Authors · 2023-11-22
> **Response to Reviewer jdbk**
>
> Dear Reviewer jdbk,
>
> We sincerely thank you for your time and effort in reviewing our paper. We value your suggestions and address them point by point in the following:
>
>
> **Q1: My largest concern lies in the validation of the experiment. The model is trained around 20000s (5.56h) at most, which is too short to validate the pretraining process since the model is far away from convergence. Notice that [1] trains model for 300B tokens, and a 160M LLM usually needs several days to converge on 16 V100s, to my knowledge. I completely understand that resource demand is high for researchers; however, it is hard for me to agree with the conclusion from the present experiment.**
>
>
> Thank you for your question and feedback. We conducted an experiment to continue pre-training the current models, as illustrated in Figure 1. As you pointed out, 'a 160M LLM usually requires several days to converge on 16 V100s,' we have decided to reserve the pretraining of larger models for future work. We are confident that our experiments, in their current form, have already substantiated that our proposed method can significantly reduce the time required for pretraining.
>
>
> **Q2: A shorter sequence surely leads to a smaller computation complexity. Therefore, this insight is not really original.**
>
> Thank you for your observation. While it's true that shorter sequences reduce computational complexity, our experiment focused on the efficiency of training without compromising the model's quality. The key contribution of our work is not just identifying the reduction in computation due to shorter sequences, but also demonstrating how our specific methodology effectively maintains, or even enhances, the model's quality within these constraints. This aspect of our research extends beyond the straightforward observation of reduced computational complexity, offering a more nuanced and practical approach to model training.
>
> Your feedback is invaluable to us, and we trust that our response has effectively addressed your concerns. In light of this, we would be grateful if the reviewer could reassess our work and possibly reconsider the score given.
>
> Thanks,\
> Authors

---

> > ### Comment · Reviewer_jdbk · 2023-11-23
> >
> > Thank you for your response. However, I would like to keep the evaluation unchanged at this stage. For a more comprehensive assessment of the proposed method's efficacy in reducing pretraining time, comprehensive empirical evidence would be beneficial. I encourage the authors to strengthen their claims with further experimental validation beyond the current assertion of confidence. Such evidence would be invaluable for a more thorough evaluation and understanding of the method's capabilities.

---

### Official Review · Reviewer_TKo2 · 2023-10-31

**Soundness:** 4 excellent
**Presentation:** 3 good
**Contribution:** 3 good
**Rating:** 6
**Confidence:** 3

**Summary:**

The paper proposes GrowLength, a pre-training strategy to progressively increase the sequence length of the training data in stages. The authors propose to use position interpolation or extrapolation to use the trained model to unseen sequence lengths. The models use relative position embeddings (ROPE) and the authors discuss the utility of the embeddings for such progressive training. On multiple model scales, the authors show the efficacy of their method with training time compared to baseline training.

**Strengths:**

The main strength of the paper lies in its easy-to-understand logic to use progressive sequence length training with ROPE embeddings. The authors take insights from position interpolation and extrapolation works and ROPE embeddings to develop the GrowLength algorithm. Furthermore, the authors clearly point out the memory and training time benefits of different input sequence lengths. Furthermore, with ablations, the authors show the algorithm's limited dependence on different progressive training schedules.

**Weaknesses:**

There are a few details that are unclear from the paper's presentation.

(a) How do the authors transition between stages? Do the authors use position interpolation or extrapolation to provide a smooth transition when the sequence length increases? Furthermore, a comparison study to a different position embedding would highlight the importance of ROPE embeddings for a smooth transition across the stages.

(b) How many sequence batches do the authors use for each stage of training? Is it proportionally set to the sequence length at each stage?

(c) Related to my second question, if the batch sizes have been changed at each stage, have the hyperparameters (Learning rate, batch size, etc.) of the baseline experiments been optimally tuned for fair comparisons?

(d) How does the improvement in perplexity relate to improvements in downstream performance? Any fine-tuned or zero-shot performance will show the general efficacy of the proposed method.

There are multiple works on efficient training that haven't been mentioned by the authors. It would be good to incorporate them to give the readers a complete view of the literature.

(1) Stacking and Layerdrop: This is a procedure to progressively increase or drop the size of the model across multiple dimensions during the course of training. [1, 2, 3, 4]. [4] had also proposed a GrowLength algorithm to incorporate into their Stacking framework.

(2) Optimization algorithm: This is a line of work that attempts to tweak the optimization algorithm to get faster pre-training. [5, 6]


1: Efficient training of bert by progressively stacking. Gong et al. 2019

2: On the transformer growth for progressive bert training. Gu et al. 2020

3: Accelerating training of transformer-based language models with progressive layer dropping. Zhang et al. 2020

4: Efficient training of language models using few-shot learning. Reddi et al. 2023.

5:  Symbolic discovery of optimization algorithms.  Chen et al. 2023

6:  A Scalable Stochastic Second-order Optimizer for Language Model Pre-training.  Liu et al. 2023

**Questions:**

Please see my questions in my previous section.

---

> ### Author Response · Authors · 2023-11-22
> **Response to Reviewer TKo2**
>
> Dear Reviewer TKo2,
>
> We sincerely thank you for your time and effort in reviewing our paper. We value your suggestions and address them point by point in the following:
>
>
> **Q(a) How do the authors transition between stages? Do the authors use position interpolation or extrapolation to provide a smooth transition when the sequence length increases? Furthermore, a comparison study to a different position embedding would highlight the importance of ROPE embeddings for a smooth transition across the stages.**
>
> We thank you for the insightful and valuable question. Regarding the stage transition, through the experimental results in Figures 1, 2, and 3, we found that directly using position extrapolation makes the transition smooth enough.
>
> For different position embeddings, we conducted the following experiments to investigate the impact of position embedding. These include various position embeddings, such as absolute positional embedding (GPT-2), and Alibi positional embedding (BLOOM).
>
>
> **Q(b) How many sequence batches do the authors use for each stage of training? Is it proportionally set to the sequence length at each stage?**
>
> We thank you for your insightful comment and would like to clarify the experimental setting. We maintained a constant product of batch_size*seq_length for all sequence lengths. Specifically, we used a batch size of 12 for a sequence length of 1024, 24 for 512, 48 for 256, and 96 for 128.
>
>
> **Q(c) Related to my second question, if the batch sizes have been changed at each stage, have the hyperparameters (Learning rate, batch size, etc.) of the baseline experiments been optimally tuned for fair comparisons?**
>
>
> Thank you for your question. We would like to explain the settings for hyperparameter tuning, such as learning rate and batch size. In large language model (LM) pretraining, it's common practice not to adjust hyperparameters, including learning rate, extensively. This is due to the unaffordable training costs. In line with this standard practice, we did not specifically tune these hyperparameters for each stage. However, we ensured that the comparisons remained as fair and consistent as possible within the constraints of our experimental setup.
>
>
>
>
> **Q(d) How does the improvement in perplexity relate to improvements in downstream performance? Any fine-tuned or zero-shot performance will show the general efficacy of the proposed method.**
>
> Thank you for your insightful comments regarding the relationship between perplexity improvements and downstream task performance. We appreciate the opportunity to further clarify and elaborate on this aspect of our work. You rightly point out the complexity in correlating perplexity improvements directly with downstream task performance. We agree that perplexity, while a valuable measure of a model's predictive capabilities, is not the sole indicator of performance in practical applications. We conducted experiments to investigate the performance comparsion on other downstream tasks. Specifically, we ran several tasks using [lm-evaluation-harness](https://github.com/EleutherAI/lm-evaluation-harness/tree/master). The results show that our proposed method outperforms the baseline in 7 out of 9 tasks.
>
> | Dataset | Staged Training | Fixed Length Training(1024) |
> |---------|---------|---------|
> | boolq   | **0.4465**  | 0.3804  |
> | multirc | **0.0189**  | 0.0105  |
> | rte     | 0.5126  | **0.5235**  |
> | wic     | **0.5235**  | 0.5047  |
> | wnli    | **0.4507**  | 0.4366  |
> | cb      | **0.5179**  | 0.4107  |
> | qasper  | **0.3356**  | 0.0     |
> | webqs   | **0.0009**  | 0.0005  |
> | wsc     | 0.3654  | 0.3654  |
>
>
> Your feedback is invaluable to us, and we trust that our response has effectively addressed your concerns. In light of this, we would be grateful if the reviewer could reassess our work and possibly reconsider the score given.
>
> Thanks,\
> Authors

---

### Meta-Review · Area_Chair_LY1P · 2023-12-06

**Metareview:**

The manuscript presents a method to accelerate the model training process by progressively feeding longer training data to change the dataLoader, which offers an effective training strategy for LLMs.

The algorithm is less dependent on the progressive training schedules [Reviewer TKo2] and the progressively increasing sequence length helps to improve accelerate training process [Reviewer jdbk, w2Jc]. With the emergence of LLMs, this is an important research question [reviewer w2Jc, gxVT]

Empirical evaluations are insufficient and the authors claims could have been strengthened through impactful validations and sufficient evaluations. Such evidence would be invaluable for a more thorough evaluation and understanding of the method's capabilities. [Reviewer TKo2]

Although the authors claim that the GrowLength presents an effective training strategy for LLMs, the method has not been demonstrated on LLMs [Reviewer gxVT]

**Justification For Why Not Higher Score:**

Empirical evaluations are insufficient and the authors claims could have been strengthened through impactful validations and sufficient evaluations. Such evidence would be invaluable for a more thorough evaluation and understanding of the method's capabilities. [Reviewer TKo2]

Although the authors claim that the GrowLength presents an effective training strategy for LLMs, the method has not been demonstrated on LLMs [Reviewer gxVT]

**Justification For Why Not Lower Score:**

N/A

---

### Decision · Program_Chairs · 2024-01-16

Reject